# Platelet-to-Lymphocyte Ratio (PLR) Is Not a Predicting Marker of Severity but of Mortality in COVID-19 Patients Admitted to the Emergency Department: A Retrospective Multicenter Study

**DOI:** 10.3390/jcm11164903

**Published:** 2022-08-21

**Authors:** Paul Simon, Pierrick Le Borgne, François Lefevbre, Lauriane Cipolat, Aline Remillon, Camille Dib, Mathieu Hoffmann, Idalie Gardeur, Jonathan Sabah, Sabrina Kepka, Pascal Bilbault, Charles-Eric Lavoignet, Laure Abensur Vuillaume

**Affiliations:** 1Emergency Department, Hôpitaux Universitaires de Strasbourg, 67000 Strasbourg, France; 2Medicine Faculty, Strasbourg University, 67000 Strasbourg, France; 3INSERM (French National Institute of Health and Medical Research), UMR 1260, Regenerative NanoMedicine (RNM), Fédération de Médecine Translationnelle (FMTS), 67000 Strasbourg, France; 4Department of Public Health, University Hospital of Strasbourg, 67000 Strasbourg, France; 5Emergency Department, Regional Hospital of Metz-Thionville, 57000 Metz, France; 6Department of Gynecologic Surgery, Hôpitaux Universitaires de Strasbourg, 67000 Strasbourg, France; 7Emergency Department, Hôpital Nord Franche Comté, 90400 Trévenans, France

**Keywords:** PLR, platelet-to-lymphocyte ratio, severity, mortality, COVID-19

## Abstract

(1) Introduction: In the present study, we investigate the prognostic value of platelet-to-lymphocyte ratio (PLR) as a marker of severity and mortality in COVID-19 infection. (2) Methods: Between 1 March and 30 April 2020, we conducted a multicenter, retrospective cohort study of patients with moderate to severe coronavirus 19 (COVID-19), all of whom were hospitalized after being admitted to the emergency department (ED). (3) Results: A total of 1035 patients were included in our study. Neither lymphocytes, platelets or PLR were associated with disease severity. Lymphocyte count was significantly lower and PLR values were significantly higher in the group of patients who died, and both were associated with mortality in the univariate analysis (OR: 0.524, 95% CI: (0.336–0.815), *p* = 0.004) and (OR: 1.001, 95% CI: (1.000–1.001), *p* = 0.042), respectively. However, the only biological parameter significantly associated with mortality in the multivariate analysis was platelet count (OR: 0.996, 95% CI: (0.996–1.000), *p* = 0.027). The best PLR value for predicting mortality in COVID-19 was 356.6 (OR: 3.793, 95% CI: (1.946–7.394), *p* < 0.001). (4) Conclusion: A high PLR value is however associated with excess mortality.

## 1. Introduction

It has been over two years that the world has been facing an unprecedented health crisis that is straining health systems. Since the first cases were identified in December 2019 in Wuhan, China [1], this pandemic caused by a new virus of the coronavirus family called SARS-CoV-2 (severe acute respiratory syndrome coronavirus 2) [2] has caused more than 6 million deaths for more than 500 million cases worldwide [3]. In France, after five waves, taking account that the first one particularly affected the Grand-East region from which our database originates, there have been over 146,000 deaths for more than 28 million cases [4]. COVID-19 (Coronavirus Disease 2019), the disease caused by this virus [2], is known mainly in its severe forms to be responsible for multi-organ failure and Acute Respiratory Distress Syndrome (ARDS). Pulmonary lesions are partly mediated by a dysregulation of the immune system leading to a “cytokine storm”, as evidenced by excessive secretion of pro-inflammatory cytokines, notably, GM-CSF, IL-6 and TNF-α [5].

It has been observed that routine laboratory tests were affected by COVID-19 with variable frequencies, such as depending on the series, 5–41.7% thrombocytopenia or up to 83% lymphopenia observed in infected patients [6,7]. Numerous studies have therefore focused on the correlation between these biological abnormalities, especially biomarkers reflecting inflammation, and the development of severe COVID-19. In addition to being associated with the development of ARDS, some biological abnormalities are also correlated with increased mortality from SARS-CoV-2: in a meta-analysis including 21 studies and finding similar results to other papers [7,8,9,10], Brandon et al. showed that significantly lower lymphocyte and platelet counts are associated with patient death compared to surviving patients, as well as increased ALT, creatinine, LDH, PT and ferritin [11]. In addition, other studies have looked more specifically at lymphopenia. Qin et al. showed in their study a significant decrease in the total number of B, T and NK cells, this decrease being even more pronounced in severe cases. T cells were shown to be more affected and they noticed a significant decrease in the regulatory T-cells subpopulation (CD3+, CD4+, CD25+, CD127low+) (9). Wang et al. obtained similar results, also finding a significant decrease in the subpopulations of CD4+ and CD8+ T cells. Here, again, the decline was even greater in severe forms, with the exception of LNK [12].

However, these biological parameters remain relatively nonspecific and the use of a combined marker, such as the platelet-to-lymphocyte ratio (PLR), could provide an additional argument to better discriminate patients at risk of developing a severe SARS-CoV-2 infection. Indeed, the PLR seems to reflect changes in platelet and lymphocyte levels related to inflammation and a pro-thrombotic state [13], which seems to be the lesioning mechanism of SARS-CoV-2. This new biomarker, which reflects inflammatory processes, is also being studied in several other types of pathologies where it provides prognostic elements or activity monitoring. In oncology, in a meta-analysis, Templeton et al. showed that a higher PLR, particularly a PLR > 185, was associated with poorer overall survival, with a pooled HR of 1.70 (95% CI, 1.47–1.95; *p* < 0.001) [14]. In a review, Gasparyan et al. summarized the research on PLR in rheumatic diseases and concluded that it can help in the diagnosis and evaluation of the activity and severity of those diseases [15]. In cardiovascular disease, Akboga et al. showed that PLR was an independent predictor of severe coronary artery disease (OR 1.043 [1.036–1.049], *p* < 0.001) and determined that an PLR > 109.5 had a sensitivity of 70%, specificity of 58% for predicting the presence of severe coronary atherosclerosis (AUC: 0.708, 95% CI: (0.68–0.73), *p* < 0.001) [16]. Meng et al. observed that in an acute non-ST-segment elevation myocardial infarction, a PLR ≥ 195.8 was significantly associated with increased 28-day mortality (HR 1.54; 95% CI: (1.09–2.18); *p* = 0.013) [17].

This study aims to assess the prognostic value of PLR in the severity and mortality of patients infected with SARS-CoV-2 on admission to the emergency department (ED).

## 2. Materials and Methods

### 2.1. Study Population and Settings

We conducted a multicentric retrospective study in six EDs in the North-East region of France. We led our study in two university hospitals (CHRU of Strasbourg in Strasbourg and CHU of Reims in Reims, France) and four general hospitals (Colmar Hospital in Colmar, Nord Franche-Comté Hospital in Belfort, Metz-Thionville Hospital in Metz and Thionville and Haguenau Hospital in Haguenau, France). These hospital centers, along with the entire Greater-East region of France, were one of the outbreak’s epicenters in Europe during the first wave.

We included all adult patients hospitalized for COVID-19 after being admitted to the ED between 1 March and 30 April 2020. All patients included in our study had a laboratory-confirmed diagnosis of COVID-19 by RT-PCR on nasopharyngeal swab (on admission). Exclusion criteria were patients who had a non-confirmed diagnosis, or those who have received outpatient care, and those who had received palliative therapy or limitation of therapeutic effort upon admission to the ED. The exclusion criteria were also patients with a medical background or treatment modifying their blood count and, therefore, their circulating lymphocyte or platelet counts (e.g., chemotherapy, immunosuppressive therapy, long- and short-term corticosteroid therapies, pre-admission antibiotic therapy, active cancer or hematological malignancies).

### 2.2. Data Collection

We retrospectively collected epidemiological, clinical and biochemical data from the patients’ electronic medical records, and standardized results in a report file. We recorded symptom onset data along with the patient’s current treatment and medical background (including cardiovascular disease, diabetes, pre-existing renal failure, cancer and hematological diseases). The primary endpoint was the prognostic value of PLR on COVID-19 mortality upon ED admission. The secondary endpoint was its prognostic value on COVID-19 severity upon ED admission. The severity of COVID-19 disease was defined by patient admission into the ICU (intensive care unit), which, during the first wave of the pandemic, was mainly associated with invasive mechanical ventilation indication. Moderate disease was defined by patient admission to conventional hospitalization units and, in fine, the requirement for simple or high-flow oxygen therapy. Ambulatory patients were excluded. Obesity was defined by a body mass index superior to 30 kg/m^2^. Standard biological parameters were collected, such as levels of creatinine, CRP, platelet count, total leukocytes and lymphocytes. Lastly, we calculated PLR values at ED admission, the ratio of platelets to circulating lymphocytes. All collected data are summarized in the Tables and Results sections.

### 2.3. Ethics

This study was approved by the local ethics committee of the University of Strasbourg in France (reference CE: 2020–39), which, in accordance with the French legislation, waived the need for informed consent of patients whose data were entirely retrospectively studied.

### 2.4. Statistical Analysis

The statistical analyses included a descriptive and an analytical section. We performed the descriptive analysis of the qualitative variables by providing the frequency of each value. We compared them in a univariate analysis by using chi-squared or Fisher’s tests in case the expected values in any of the cells of a contingency table was below 5. We performed the descriptive analysis of the continuous variables by providing median, and first and third quartiles of each value. We compared them in an univariate analysis by using a non-parametric Mann–Whitney test or, in case the variables followed a normal distribution by using a Welsh’s test. Using statistically significant results obtained from univariate analyses and clinically relevant variables, a multivariate logistic model was performed to assess disease severity, then the in-hospital mortality. A backward stepwise method was performed. Receiver operating characteristic (ROC) curves were constructed and the best cut-off value of PLR discriminating severe from moderate patients, and patients who died during their stay from those who survived were determined by using the Yunden’s index. *p*-values < 0.05 were considered significant and the confidence interval (CI) was 95%. Analyses were performed with the R software in version 4.0.2 (R Core Team 2020. R: A language and environment for statistical computing. R Foundation for Statistical Computing, Vienna, Austria), as well as with all the software packages required to conduct the analysis.

## 3. Results

### 3.1. Characteristics of the Study Population

During the study period, a total of 49,326 patients were admitted to the EDs of all six hospitals. Of these patients, 4470 had a laboratory-confirmed SARS-CoV-2 infection, 1685 received ambulatory care, 1750 met the exclusion criteria and, in fine, 1035 patients were included in our study (flowchart: Figure 1).

Our cohort had a median age of 69 (58.0–79.0) years and was predominately male (58.8%). Regarding the comorbidities, 1/3 of our study population was obese (36.9%) and had coronary heart disease (34.5%), over 1/2 of the patients (56.7%) had hypertension, over 1/4 of them (26.6%) had a history of diabetes and 23.2% of them presented chronic kidney disease. Only 5.4% had chronic obstructive lung disease. At admission, even if the median lymphocyte count was significantly lower in the group presenting severe COVID-19 compared to that with moderate COVID-19 (0.780, 95% CI: (0.590–1.122)) vs. 0.900, 95% CI: (0.640–1.220), *p* = 0.003)), we did not find a significant difference between these two groups for platelet count and PLR. The main clinical and biochemical patient characteristics are summarized in Table 1.

We performed the analysis to observe if there was any interest in carrying on our study and if there was a difference in PLR between living and dead patients. The median PLR was significantly higher in the group of patients who died compared to those who survived (242.3, 95% CI: (164.6–385.7) vs. 221.4, 95% CI: (154.7–319.4), *p* = 0.043)); Table 2.

### 3.2. Biochemical Factors Associated and Factors Predicting COVID-19 Severity

Of the entire study population, 789 patients (76.2%) had moderate disease, whereas 246 (23.8%) had severe disease requiring ICU management. When comparing these two subgroups, age (70 vs. 66 years, *p* < 0.001) and gender (*p* < 0.001) differed significantly. Patients admitted to the ICU had fewer cardiovascular (*p* = 0.004) and renal (*p* = 0.002) comorbidities. In the multivariate analysis, adjusted for age, gender, complications, and laboratory findings following a backward stepwise selection, none of the parameters studied were associated with the severity of infection. The results are summarized in Table 3.

We made a receiver operating characteristics (ROC) curve to predict the risk of disease severity. Regarding the PLR during admission, the area under the curve (AUC) was 0.54 (95% CI: (0.497–0.582)). The best cut-off for predicting the risk of infection severity was 369.7; it yielded a sensitivity of 23.8% (95% CI: (18.6–29.7)) and specificity of 83.8% (95% CI: (81.0–86.3)). In the multivariate analysis, if PLR was greater than 369.7, the OR was valued at 1.884 (95% CI: (1.130–140), *p* = 0.015) (Figure 2).

### 3.3. Biochemical Factors Associated and Factors Predicting COVID-19 Mortality

Mortality analysis included 1023 patients, as 12 patients (1.2%) were lost to follow-up. A total of 139 patients died during their hospital stay, representing 13.6% of our cohort, while 884 (86.4%) survived. Non-surviving patients were significantly older (78 versus 67 years, *p* < 0.001). They were more likely to have a medical history of hypertension (*p* < 0.001), chronic obstructive lung disease (*p* < 0.001), chronic kidney disease (*p* < 0.001) and coronary heart disease (*p* < 0.001). Biochemically, higher levels of creatinine (96.0, 95% CI: (77.5–144.5) vs. 76.0, 95% CI: (62.0–94.0), *p* < 0.001), CRP (100.0, 95% CI: (56.0–158.0) vs. 78.5, 95% CI: (37.0–139.0), *p* = 0.008) and lactate (1.4, 95% CI: (1.1–1.9) vs. 1.2, 95% CI: (0.9–1.5), *p* < 0.001) were observed in the non-surviving subgroup. Regarding the cell blood count, lymphopenia was more profound (0.720, 95% CI: (0.500–1.000) vs. 0.890, 95% CI: (0.650–1.220), *p* < 0.001), and platelet count was significantly lower (181.0, 95% CI: (138.3–246.0) vs. 196.0, 95% CI: (153.3–248.0), *p* = 0.031) in the non-surviving subgroup. Upon admission to the ED, the lymphocyte count was significantly lower and PLR values were significantly higher, and both were associated with mortality in univariate analysis (respectively, *p* = 0.004 and *p* = 0.042). However, the only biochemical parameter significantly associated with mortality in multivariate analysis was the platelet count (OR: 0.996, 95% CI: (0.996–1.000), *p* = 0.027). These results are summarized in Table 4.

We created receiver operating characteristics (ROC) curves to predict the risk of disease mortality. Regarding the PLR at admission, the area under the curve (AUC) was 0.55 (95% CI: (0.498–0.611)). The best cut-off for predicting the risk of infection severity was 356.6: it yielded a sensibility of 34.3% (95% CI: (26.3–43.0)) and a specificity of 82.4% (95% CI: (79.7–84.9)). In the multivariate analysis, if the PLR was greater than 356.6, the OR was valued at 3.793 (95% CI: (1.946–7.394), *p* < 0.001) (Figure 3).

## 4. Discussion

The aim of our study was to investigate the prognostic value of the PLR in a cohort of SARS-CoV-2-infected patients from their admission to the ED. We selected our patients as carefully as possible to limit confounding factors that could have altered the CBC. Our study did not show significant results to recognize PLR as an efficient marker to discriminate among hospitalized patients, those likely to develop a severe form and requiring ICU management. Furthermore, in the univariate analysis, the PLR was significantly higher in patients who died compared to those who survived, although this association was not observed in the multivariate analysis.

The results of the studies investigating PLR in COVID-19 remain heterogeneous. Some studies seem to indicate that the PLR would be an efficient predictive marker of the severity and mortality of COVID-19, which slightly differs according to our results. In addition to the limited number of subjects, which may be responsible for the lack of power presented in these studies and explain the difference of results, it is important to notice that the inclusion criteria were also different in other studies. Indeed, some authors used a control group of COVID-19-negative patients as a comparison [18,19]. Others, including a majority of the studies from China, had inclusion criteria that referred to the criteria met from the General Office of the National Health Commission of China [20] and classified patients into two subgroups: “mild” and “common", which, in our study, met the exclusion criteria [21,22,23]. Finally, the PLR was higher and appeared to be a valuable marker to discriminate non-infected patients from infected patients, such as patients with pauci-symptomatic forms of the disease from patients meeting the hospitalization criteria. However, the PLR did not make it possible to discriminate between moderate and severe forms requiring intensive care management. Wang R et al., in a study with a patient selection similar to ours, were one of the few researchers to obtain results for the PLR that were similar to our study. The PLR was significatively higher in the non-surviving group compared to the surviving group (237.32, CI 95%: (160.15–400.96) vs. 173.29, CI 95%: (132.35–252.22), *p* < 0.001), and in the univariate analysis there was an association between a high PLR and mortality (OR: 1.004, CI 95%: (1.002–1.05), *p* < 0.001), which they did not observe in the multivariate analysis (OR: 1.003, CI 95%: (0.999–1.007), *p* = 0.154) [24]. 

Another particularity of our study was its exclusion criteria. Indeed, we decided to exclude patients with comorbidities or treatments that could alter the CBC and modify the PLR. However, there were some situations where the PLR was often increased, with a negative correlation with the underlying disease, and which had a poor prognosis in the case of SARS-CoV-2 infection. Solid cancers, for example, have been described as a risk factor for mortality in COVID-19 [10,25] cases and a high PLR was associated with advanced disease and mortality [14,26]. Similarly, we excluded patients who received corticosteroid therapy, which tends to increase the platelet count and to a lesser extent the lymphocyte count [27], and thus increase the PLR. However, corticosteroid therapy is now an integral part of the treatment used for COVID-19 [28].

From a pathophysiological point of view, several causes are put forward to explain lymphopenia and thrombocytopenia. Lymphopenia could be linked to a cell-exhaustion phenomenon, direct viral lymphocyte infection, bone marrow infection, apoptosis of lymphocytes led by inflammation and inhibition of lymphocytes by metabolic dysregulation [29,30]. Thrombocytopenia is essentially linked to platelet consumption, but also to lower levels of production due to bone marrow damage and to an immunological phenomena leading to platelet destruction [31]. However, since those do not appear from the same mechanism, the onset speed of these two biological anomalies differs. Studies have analyzed the evolution of these biological abnormalities during infection. Indeed, there is high variability over time concerning the number of platelets and lymphocytes, particularly during the early onset of the disease [32,33,34,35]. This also raises the question of PLR’s variability and its interpretation. According to our results, we can also ask ourselves about the utility of this ratio and, more broadly, the contribution of biological ratios that have emerged in recent years, such as the neutrophil-to-lymphocyte ratio (NLR), the lymphocyte-to-monocyte ratio (LMR) or the systemic immune-inflammation index (SII). In their study, Pierrakos et al. reviewed the emergence of numerous new inflammatory markers and pointed out that the majority were evaluated by less than five studies and even more were evaluated by studies with small numbers, or that these studies answered a specific clinical question rather than addressing their general diagnostic or prognostic properties [36]. A state of play seems to be necessary for these new ratios as biomarkers of inflammation.

### Limitations

Firstly, this was a retrospective study, which means that although we added a number of exclusion criteria (including comorbidities that alter blood cell count and therefore the number of circulating lymphocytes and platelets), the data were subject to further confounding factors.

With the aim of keeping away the risk of confounding factors, we excluded patients who were potentially more severe and more frequently hospitalized. Similarly, we did not take into account patients who received ambulatory care. In fact, we probably minimized the effects studied, the same way that we cannot, with our results, discriminate between patients with “mild and common”, “moderate” and “severe” forms.

Finally, our patients were exclusively included in the first wave, which had several implications. Firstly, many patients received non-recommended treatments before their admission to the ED, such as antibiotics—which could modify the CBC and which we excluded—or antimalarials. Secondly, patient management has evolved considerably since the first wave, notably with the widespread use of anticoagulants to prevent thromboembolic events and corticoids, which can alter the CBC. Finally, the disease itself has evolved with the emergence of new variants, the most recent of which appear to be more contagious but less virulent.

## 5. Conclusions

Although the PLR is an interesting marker of inflammation, it does not appear to be a good prognostic marker to discriminate the most severe patients infected with SARS-CoV2 admitted to an ED. A high PLR could, however, be associated with excess mortality. Further studies would be needed to confirm this.

## Figures and Tables

**Figure 1 jcm-11-04903-f001:**
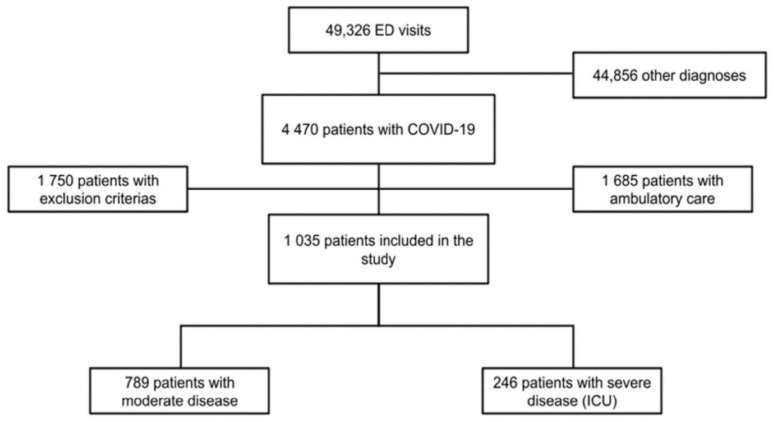
Flowchart of the study. Legend: ED: emergency department, ICU: intensive care unit, COVID-19: coronavirus disease.

**Figure 2 jcm-11-04903-f002:**
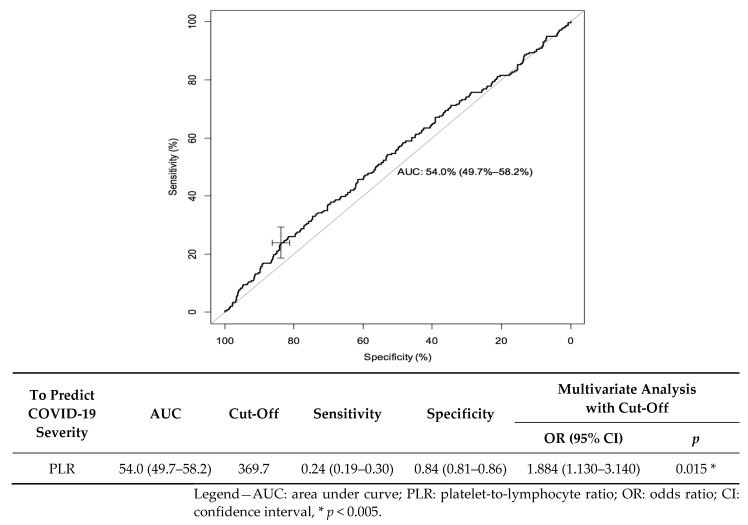
Receiver operating characteristics (ROC) curve for PLR as a predictive factor of severe COVID-19.

**Figure 3 jcm-11-04903-f003:**
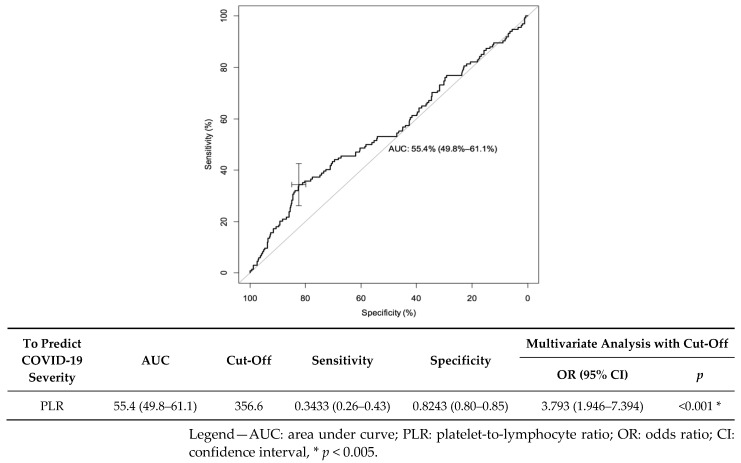
Receiver operating characteristics (ROC) curve for PLR as a predictive factor of COVID-19 mortality.

**Table 1 jcm-11-04903-t001:** General characteristics, comorbidities and laboratory findings of study population with moderate and severe COVID-19.

		All Patients	Moderate COVID-19	Severe COVID-19	*p*
		(n = 1035)	(n = 789)	(n = 246)	
Characteristics					
Age (years)		69.0 (58.0–79.0)	70.0 (58.0–81.0)	66.0 (57.3–72.0)	<0.001 *
Gender male		609 (58.8)	433 (54.9)	176 (71.5)	<0.001 *
Current smoker		46 (4.4)	34 (4.3)	12 (4.9)	0.706
Comorbidities					
Hypertension		587 (56.7)	453 (57.4)	134 (54.5)	0.416
Diabetes		275 (26.6)	202 (25.6)	73 (29.7)	0.207
Obesity	BMI (kg/m^2^) (30, 40)	253 (33.2)	172 (31.2)	81 (38.6)	0.056
	BMI (kg/m^2^) ≥ 40	28 (3.7)	21 (3.8)	7 (3.3)	0.966
COPD		56 (5.4)	44 (5.6)	12 (4.9)	0.672
Chronic kidney disease		237 (23.2)	199 (25.5)	38 (15.8)	0.002 *
Coronary heart disease		357 (34.5)	291 (36.9)	66 (26.8)	0.004 *
Laboratory findings					
Lymphocyte count, ×10^9^ per L		0.870 (0.630–1.200)	0.900 (0.640–1.220)	0.780. (0.590–1.122)	0.003 *
Platelet count, ×10^9^ per L		194.5 (152.0–248.0)	196.0 (154.0–247.0)	192.0 (144.0–253.0)	0.518
PLR		223.3 (156.5–329.0)	219.9 (154.7–320.5)	238.5 (162.4–357.7)	0.061
Outcomes					
Mortality		139 (13.6)	82 (10.4)	57 (24.1)	<0.001 *
Length of hospital stay (days)		10.0 (7.0–17.3)	8.0 (6.0–12.0)	24.0 (17.0–38.0)	<0.001 *

Data are all expressed in median (Q1–Q3) or *n* (%) where *n* is the total number of patients with available data. * *p* < 0.05. Legend: BMI: body mass index, COPD: chronic obstructive pulmonary disease, PLR: platelet-to-lymphocyte ratio.

**Table 2 jcm-11-04903-t002:** General characteristics, comorbidities and laboratory findings of the study population, surviving and dying from COVID-19.

		All Patients	Survivor	Non Survivor	*p*
		(n = 1035)	(n = 884)	(n = 139)	
Characteristics					
Age (years)		69.0 (58.0–79.0)	67.0 (56.0–77.0)	78.0 (70.0–86.0)	<0.001 *
Gender male		609 (58.8)	517 (58.5)	85 (61.2)	0.553
Current smoker		46 (4.4)	42 (4.8)	4 (2.9)	0.322
Comorbidities					
Hypertension		587 (56.7)	477 (54.0)	103 (74.1)	<0.001 *
Diabetes		275 (26.6)	227 (25.7)	42 (30.2)	0.259
Obesity	BMI (30, 40)	253 (33.2)	222 (33.1)	30 (34.9)	0.889
	BMI ≥ 40	28 (3.7)	27 (4.0)	1 (1.2)	0.191
COPD		56 (5.4)	38 (4.3)	18 (13.0)	<0.001 *
Chronic kidney disease		237 (23.2)	189 (21.6)	47 (35.3)	<0.001 *
Coronary heart disease		357 (34.5)	283 (32.1)	72 (51.8)	<0.001 *
Laboratory findings					
Lymphocyte count, ×10^9^ per L		0.870 (0.630–1.200)	0.890 (0.650–1.220)	0.720 (0.500–1.000)	<0.001 *
Platelet count, ×10^9^ per L		194.5 (152.0–248.0)	196.0 (153.3–248.0)	181.0 (138.3–246.0)	0.031 *
PLR		223.3 (156.5–329.0)	221.4 (154.7–319.4)	242.3 (164.6–385.7)	0.043 *
C-reactive protein, mg/L		81.0 (39.0–142.0)	78.5 (37.0–139.0)	100.0 (56.0–158.0)	0.008 *
Creatinine, μmol/L		78.0 (64.0–98.0)	76.0 [62.0–94.0)	96.0 (77.5–144.5)	<0.001 *
Lactate, mmol/l		1.2 (0.9–1.6)	1.2 (0.9–1.5)	1.4 (1.1–1.9)	<0.001 *

Data are all expressed in median (Q1–Q3) or *n* (%) where *n* is the total number of patients with available data. * *p* < 0.05. Legend: BMI: body mass index, COPD: chronic obstructive pulmonary disease, PLR: platelet-to-lymphocyte ratio.

**Table 3 jcm-11-04903-t003:** Univariate and multivariate analyses for risk factor for COVID-19 severity.

	All Patients	Moderate COVID-19	Severe COVID-19	Univariate Analysis		Multivariate Analysis	
				OR (95% CI)	*p*	OR (95% CI)	*p*
Lymphocyte count, 10^9^ per L	0.870 (0.630–1.200)	0.900 (0.640–1.220)	0.780. (0.590–1.122)	0.827 (0.616–1.110)	0.206	0.937 (0.647–1.357)	0.729
Platelet count, 10^9^ per L	194.5 (152.0–248.0)	196.0 (154.0–247.0)	192.0 (144.0–253.0)	1.000 (0.998–1.002)	0.979	1.000 (0.998–1.003)	0.765
PLR	223.3 (156.5–329.0)	219.9 (154.7–320.5)	238.5 (162.4–357.7)	1.000 (1.000–1.001)	0.107	1.001 (1.000–1.001)	0.107

Data are all expressed in median (Q1–Q3) or n (%) where n is the total number of patients with available data. Legend: OR: odds ratio, CI: confidence interval, PLR: platelet-to-lymphocyte ratio.

**Table 4 jcm-11-04903-t004:** Univariate and multivariate analyses for risk factors for COVID-19 mortality.

	All Patients	Survivor	Non Survivor	Univariate Analysis		Multivariate Analysis	
				OR (95% CI)	*p*	OR (95% CI)	*p*
Lymphocyte count, 10^9^ per L	0.870 (0.630–1.200)	0.890 (0.650–1.220)	0.720 (0.500–1.000)	0.524 (0.336–0.815)	0.004 *	0.756 (0.393–1.456)	0.403
Platelet count, ×10^9^ per L	194.5 (152.0–248.0)	196.0 (153.3–248.0)	181.0 (138.3–246.0)	0.998 (0.995–1.000)	0.078	0.996 (0.992–1.000)	0.027 *
PLR	223.3 (156.5–329.0)	221.4 (154.7–319.4)	242.3 (164.6–385.7)	1.001 (1.000–1.001)	0.042 *	1.000 (0.999–1.002)	0.444

Data are all expressed in median (Q1–Q3) or *n* (%) where *n* is the total number of patients with available data. * *p* < 0.05. Legend: OR: odds ratio, CI: confidence interval, PLR: platelet-to-lymphocyte ratio.

## Data Availability

All data analyzed as part of the study are included.

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
