# Peer review of "Platelet-to-Lymphocyte Ratio (PLR) Is Not a Predicting Marker of Severity but of Mortality in COVID-19 Patients Admitted to the Emergency Department: A Retrospective Multicenter Study"

_jcm, 2022, doi:10.3390/jcm11164903_

Round 1

Reviewer 1 Report

The manuscript “Platelet-to-lymphocyte ratio (PLR) as a predicting marker of severity and 2 mortality in the Emergency Department patients with SARS-CoV-2: a retrospective multicenter study.”, describes the findings of a multicenter, retrospective cohort study looking at close to 1000 patients that were admitted to the emergency department. No association of PLR with severe disease was detected, but lymphocyte count and PLR were associated with mortality. The authors conclude that a high PLR is associated with mortality, although with several conflicting results from work form other groups. I find the study clinically useful addressing an important question which is how to assess severity of COVID early on in the disease progression.

Major comments:

-          I would change the title as to reflect the results. I would suggest: “Platelet-to-lymphocyte ratio (PLR) is not a predicting marker of severity but of mortality in COVID-19 patients admitted to the Emergency Department: a retrospective multicenter study”.

-          Exclusion criteria should be further explained since this is not clear in the M&M and then it is an important piece of the discussion. Please explain the exclusion criteria that affect the cbc, since chronic diseases such as the ones in the comorbidities (and included in the study) might also affect cdc . “In our study we decided to exclude patients with comorbidities or treatments that 243 could alter the CBC and modify the PLR”

Minor comments:

The introduction should state to further detail other diseases that might affect the PLR.

Author Response

Comments and Suggestions for Authors

The manuscript “Platelet-to-lymphocyte ratio (PLR) as a predicting marker of severity and 2 mortality in the Emergency Department patients with SARS-CoV-2: a retrospective multicenter study.”, describes the findings of a multicenter, retrospective cohort study looking at close to 1000 patients that were admitted to the emergency department. No association of PLR with severe disease was detected, but lymphocyte count and PLR were associated with mortality. The authors conclude that a high PLR is associated with mortality, although with several conflicting results from work form other groups. I find the study clinically useful addressing an important question which is how to assess severity of COVID early on in the disease progression.

Major comments:

 -          I would change the title as to reflect the results. I would suggest: “Platelet-to-lymphocyte ratio (PLR) is not a predicting marker of severity but of mortality in COVID-19 patients admitted to the Emergency Department: a retrospective multicenter study”.

Response: We thank the reviewer for this powerful title. We fully follow his advice.

-          Exclusion criteria should be further explained since this is not clear in the M&M and then it is an important piece of the discussion. Please explain the exclusion criteria that affect the cbc, since chronic diseases such as the ones in the comorbidities (and included in the study) might also affect cdc . “In our study we decided to exclude patients with comorbidities or treatments that 243 could alter the CBC and modify the PLR”

Response: We had this sentence in the method section to explain which patients were excluded “ Patients with a medical background or treatment modifying their blood count and, therefore, their circulating lymphocyte count or platelet count (e.g., chemotherapy, immunosuppressive therapy, long- and short-term corticosteroid therapy, pre-admission antibiotic therapy, active cancer, or hematological malignancies) were also excluded from our study.

We made it clear in the manuscript that these were the exclusion criteria for the study.

Minor comments:

The introduction should state to further detail other diseases that might affect the PLR.

Response: We have taken this excellent advice and detailed these elements in the introduction. We hope that this revised version will meet your requirements.

Reviewer 2 Report

Dear Authors,

thank you for the possibility to review your interesting research.

In my opinion, the article is well and clearly written. The results are presented correctly. Below you will find some of my suggestions.

Line 58  typo error - should be SARS-CoV2 not SAR-CoV2

Line 66 - repetition in the text - found - finding - consider the use of "Wang et al presented or reported or cane to a conclusion

Line 88  We included all adult hospitalized patients for COVID 19 - I suggest to change the order orf the sentence: We included all adult patients hospitalized for COVID 19. Please, consider.

Line 110 - Could you please specify when or in what phase of the disease the lab tests were performed? On admission?

Table 1 - measurement units should be written in the table. In most cases you wrote whem but without "days" in "length of horpital stay and "mortality"

Table 2 - the same for BMI (kg/m2)

Line 181 - Please consider writing the full sentence instead of just Figure 2. For example: the results are shown in Figure 2. The same for Figure 3.

Figure 2 - *=p<0.005

Line 215 - patients are not infected with COVID-19 but they develop COVID19 in answer to the SARSCoV2 infection. Please, clarify.

Line 235 - I would propose o wtite "patients meeting the hospitalization criteria"

References: please, standarize the references: e.g. lines 318, 321 (disponible sur), deletion of the names of the months of publication of articles, e.g. lines 374, 376 (dec, mai).

 I have no other objections. In my opinion this research is worth to publish after these minor corrections have been made.

Good luck.

Author Response

Dear Authors,

thank you for the possibility to review your interesting research.

In my opinion, the article is well and clearly written. The results are presented correctly. Below you will find some of my suggestions.

Line 58  typo error - should be SARS-CoV2 not SAR-CoV2

Response: Done

Line 66 - repetition in the text - found - finding - consider the use of "Wang et al presented or reported or cane to a conclusion

Response: Done

Line 88  We included all adult hospitalized patients for COVID 19 - I suggest to change the order orf the sentence: We included all adult patients hospitalized for COVID 19. Please, consider.

Response: Done

Line 110 - Could you please specify when or in what phase of the disease the lab tests were performed? On admission?

Response: Yes, on admission. Done!

Table 1 - measurement units should be written in the table. In most cases you wrote whem but without "days" in "length of horpital stay and "mortality"

Response: We thank the reviewer for his attention. Hospital stay in in days. Mortality is in absolute number.

Table 2 - the same for BMI (kg/m2)

Line 181 - Please consider writing the full sentence instead of just Figure 2. For example: the results are shown in Figure 2. The same for Figure 3.

 Response: Done

Figure 2 - *=p<0.005

 Response: Done

Line 215 - patients are not infected with COVID-19 but they develop COVID19 in answer to the SARSCoV2 infection. Please, clarify.

  Response: Done

Line 235 - I would propose o wtite "patients meeting the hospitalization criteria"

 Response: Done

References: please, standarize the references: e.g. lines 318, 321 (disponible sur), deletion of the names of the months of publication of articles, e.g. lines 374, 376 (dec, mai).

 Response: Done

 I have no other objections. In my opinion this research is worth to publish after these minor corrections have been made.

Good luck.

 Response : We thank the reviewer for the time he took for our work. We hope that the revised version will meet his expectations.
